# Enteral Nutrition in Patients with Inflammatory Bowel Disease. Systematic Review, Meta-Analysis, and Meta-Regression

**DOI:** 10.3390/nu11112657

**Published:** 2019-11-04

**Authors:** Jose M. Comeche, Pablo Caballero, Ana Gutierrez-Hervas, Sofia García-Sanjuan, Iris Comino, Cesare Altavilla, Jose Tuells

**Affiliations:** 1Department of Community Nursing, Preventive Medicine and Public Health and History of Science (SPAIN), University of Alicante, San Vicente del Raspeig, 03690 Alicante, Spain; josemiguelcomeche@gmail.com (J.M.C.); iriscomino@gmail.com (I.C.); eatingfaster@gmail.com (C.A.); tuells@ua.es (J.T.); 2Department of Nursing (SPAIN), University of Alicante, San Vicente del Raspeig, 03690 Alicante, Spain; ana.gutierrez@ua.es (A.G.-H.); sofia.garcia@ua.es (S.G.-S.)

**Keywords:** inflammatory bowel diseases, enteral nutrition, systematic review, meta-analysis, Crohn’s disease

## Abstract

Inflammatory bowel disease (IBD) is a chronic disease mediated by the immune system and is characterized by inflammation of the gastrointestinal tract. One of the possible treatments for this pathology is a change in the type of diet, of which enteral nutrition (EN) is one. This study is to understand how the use of EN can affect the adult population diagnosed with IBD. We conducted a systematic review, meta-analysis, and a meta-regression. On the different databases (MEDLINE, Scopus, Cochrane, LILACS, CINAHL, WOS), we found 363 registers with an accuracy of 12% (44 registers). After a full-text review, only 30 research studies were selected for qualitative synthesis and 11 for meta-analysis and meta-regression. The variables used were Crohn’s disease activity index (CDAI), C-reactive protein (CRP), and erythrocyte sedimentation rate (ESR). EN has been shown to have efficacy for the treatment of Crohn’s disease and is compatible with other medicines. As for the CDAI or rates of remission, there were no differences between enteral and parenteral nutrition. Polymeric formulas have shown better results with respect to the CRP. The long-term treatment could dilute the good CDAI results that are obtained at the start of the EN treatment.

## 1. Introduction

Inflammatory bowel disease (IBD) is a chronic disease mediated by the immune system and characterized by the inflammation of the gastrointestinal tract. IBD includes Crohn’s disease (CD) as well as ulcerative colitis (UC) [1]. UC affects the large intestine and is generally observed as a superficial ulcer due to an inflammatory reaction localized to the mucosa and the submucosa. However, CD occurs all along the intestinal tract (from mouth to anus) and involves the entire intestinal layer [2].

The prevalence and incidence of IBD has increased worldwide and is increasingly diagnosed in young individuals [3]. As it is a chronic, incurable, and low-mortality disease, it is expected that the decrease of the global burden of the disease in the next decade will require a two-pronged solution that implies research on prevention interventions as well as innovations in the care of these patients [3,4].

The etiology of IBD is still greatly unknown, and recent evidence indicates that the genetic susceptibility of the individual, the environment, the intestinal microbial flora, and the immune responses are all factors that are involved and functionally integrated in the pathogenesis of IBD [5]. IBD can provoke various symptoms that include abdominal pain, low fever, fatigue, weight loss, diarrhea, bloody feces, etc. [6].

Within the identification of the environmental risk factors, diet is one of the most important as it regulates intestinal inflammation by modifying the intestinal microbiota, which has an effect on the gastrointestinal permeability [7,8]. Therefore, it can induce the expression of disease genes and determine the cell’s phenotype and function in IBD [7,8]. One of the possible treatments for this pathology is a change in the type of diet [9]. 

One of the potential changes in diet is the use of enteral nutrition (EN), which is based on the administration of enteral foods/formulas through different means. These foods are nutritionally-complete liquid mixtures of pre-digested foods that have carbohydrates such as simple sugars, fats such as different types of oils, and nitrogen as protein, along with vitamins and minerals [10]. Within the elemental formulas, different classes can be distinguished as a function of the nitrogen source: elemental formulas are based on amino acids, semi-elemental formulas are based on oligo-peptides, and polymeric formulas are based on whole proteins [11]. 

Diverse authors have highlighted that EN, especially in the form of exclusive enteral nutrition (EEN), is a type of therapy established to induce the remission of CD in the infant population, although its role as a first line therapy for CD in adults has not been defined yet and its mechanism of action for palliating the symptoms of IBD is not completely understood [9,12]. Authors such as Guagnozzi et al. suggest that the interaction between the composition of specific dietary formulas or nutrients and IBD should be investigated to add new knowledge to the etiopathogenesis of the disease in nutritional intervention [13]. 

Therefore, the main objective of this study was to understand how the use of EN can affect the adult population diagnosed with IBD.

## 2. Materials and Methods

To achieve this objective, a systematic review was conducted in agreement with the procedures and verification list described by PRISMA [14]. Afterward, a meta-analysis on the more common results and a meta-regression with the co-variables, type of enteral nutrition, and period of treatment, were conducted.

### 2.1. Systematic Review

A search of scientific works was conducted in the MEDLINE database through the system of open retrieval system on the Internet such as PubMed, Cochrane, Scopus, Web of Science, CINAHL, and LILACS. Studies conducted over time up to 5 January 2019 were compiled.

#### 2.1.1. Inclusion and Exclusion Criteria

The studies selected had to comply with the following inclusion criteria: refer to an adult population (older than 18) diagnosed with some type of IBD; study the effect of enteral nutrition within IBD; were clinical trials; and in the English, Spanish, Portuguese, French, or German languages.

The following articles were excluded: those that referred to the infant population; to animals, to the use of EN in a healthy adult population; those that sought the effect of oral exclusion diets on IBD; observational studies; and those based on secondary sources.

#### 2.1.2. Search Equation

To include content linked to the intervention EN, a specific descriptor was used (MeSH) such as “Enteral Nutrition”, and the term “Enteral Nutrition” in the title or abstract.

For the content linked to the population, we utilized the descriptor that referred to the disease “Inflammatory bowel diseases”, and its equivalent term in the title or abstract.

Additionally, the filters “Humans”, “Adult”, and “Clinical Trial” were utilized to achieve our objective. 

Therefore, the main search equation designed for this study was:

((“Inflammatory Bowel Diseases”[Mesh] OR “Inflammatory Bowel Diseases”[Title/Abstract]) AND (“Enteral Nutrition”[Mesh] OR “Enteral Nutrition”[Title/Abstract])) AND (Clinical Trial[ptyp] AND Humans[Mesh] AND adult[MeSH]) 

The search equation was adapted to each and all of the databases described previously. The process was conducted between the months of May and June, 2019.

#### 2.1.3. Selection Process

After eliminating the duplicate records, the process of selection was conducted in two phases. The first consisted of reviewing the titles and abstracts of all the article records that resulted from the adapted search equations and were shown by the databases by using the inclusion and exclusion criteria and the objective of the study as the screening measure. The screening and selection of the records/articles were conducted independently by the two researchers, both experts in the fields of nutrition. These researchers agreed on the discrepancies found in order to define the final suitability of the records/articles found in the databases. The precision of the search was calculated, based on the ratio of the full-text articles selected for the review, divided by the number of records found by the search equation, and multiplied by one hundred.

The second phase was conducted by applying the inclusion/exclusion criteria to the complete texts of all of the scientific studies selected in the first phase, thus ensuring the relevance of each one. In order to obtain studies that were not accessible via the Internet, we used three methods: Researchgate, the correspondence author, and interlibrary loan. Only three were recovered through interlibrary loan.

#### 2.1.4. Evaluation of the Quality of the Studies

The evaluation of the methodological quality of the included studies was performed by two independent researchers by using the Cochrane risk of bias tool (RoB 2) for clinical trials [15]. This tool is structured into five domains through which bias might be introduced into the result: the randomization process, deviations from intended interventions, missing outcome data, measurement of the outcome, and selection of the reported results. For each study, the response options for an overall risk-of-bias judgement were “low risk of bias”, “some concerns”, and “high risk of bias”.

### 2.2. Meta-Analysis and Meta-Regression

To calculate the effect size of the enteral nutrition on the variables: the Crohn’s disease activity index (CDAI), C-reactive protein (CRP), and erythrocyte sedimentation rate (ESR), a meta-analysis, were performed. For this, the model of fixed effects and the model of random effects were utilized. The results were presented as a forest-plot, along with the percent heterogeneity and its confidence interval at 95%, the T value, and the heterogeneity test.

To explore the influence of each study over the effect size, we used a leave-one-out method; pooled estimates were calculated by omitting one study at a time. In addition, we plotted a scatter plot introduced by Baujat et al. [16] On the *x*-axis, the contribution of each study to the overall heterogeneity statistic was plotted. On the *y*-axis, the standardized difference of the overall treatment effect with and without each study was plotted; this quantity describes the influence of each study on the overall treatment effect. Therefore, studies that fell in the top right quadrant of the Baujat plot had the most influence. 

Publication bias occurs when only favorable results are published, and this could have consequences on the results of the meta-analyses if these were included. To analyze the publication bias, a non-parametric analysis was conducted as proposed by Duval and Tweedie [17], based on the funnel-plot, estimating and adjusting for the number and outcomes of missing studies in the meta-analysis. Another less-conservative proposal to estimate the number and outcomes of missing studies was proposed by Copas et al. [18]. 

The meta-regression was utilized to understand if the type of enteral nutrition (polymeric or elemental), age (years), or the duration of the intervention (days) modified the effect size of the resulting variables CDAI, CRP, and ESR as a function of the type of nutrition. All the calculations were conducted within an R programming environment by utilizing the packages meta version 4.9-6 [19] and metasens version 0.4-0 [20].

## 3. Results

### 3.1. Systematic Review

As a result of the specific search equations used on the different databases, a total of 438 records of scientific articles were found. A total of 75 records were duplicated, leaving a total of 363 records without duplication. In the first phase of the study, exactly 319 study records were discarded, leaving 44 full-text studies to review, so the accuracy was 12%. The reasons for not including them were that 131 records showed that the study utilized a design that was not adequate, 100 did not use an adult population, 50 did not study the effect of EN, three were written in another language other than the ones cited above, (two in Japanese and one in Chinese), 12 did not refer to humans, six did not refer to the IBD, and 17 were conducted without showing results (Figure 1).

In the second phase, seven studies were not utilized as they could not be obtained in electronic format, not even after using the methods explained in the methodology. In addition, seven trials were removed, one for being written in Turkish, another due to defects in its design, another for not studying the effects of EN, and four because the population studied was not diagnosed with IBD. Therefore, only 30 research studies [10,21,22,23,24,25,26,27,28,29,30,31,32,33,34,35,36,37,38,39,40,41,42,43,44,45,46,47,48,49] were selected, as shown in Figure 1.

As for the designs of the studies included, 16 controlled and randomized clinical studies (53.3%), nine non-randomized, controlled clinical trials (30%), and five non-randomized, non-controlled clinical trials (16.7%) were found. In addition, 28 of the studies found showed results that specifically referred to CD, and two studies had results on UC and CD, under the category of IBD. Additionally, 23 studies mentioned results of the disease in its active form, four studies in the shape of remission, and the rest did not indicate any. Figure 2 shows this information in a chronological manner.

As for the variety of the types of formulas employed, 19 studies utilized an elemental formula, 11 studies utilized a polymeric formula, two studies a semi-elemental formula, and three studies a type of parenteral nutrition (PN). Likewise, it should be mentioned that various types of formulas were often used in the same study. Thus, the following commercial formulas were employed: “E028”, “Novasource”, “Peptisorb”, “Elental”, “E028 Extra”, “Vivonex-TEN”, “Peptison”, “Peptamen”, “Vivonex HN”, “Realmentyl”, “Triosrbon”, “Vital”, “Vivonex”, “Fortison”, “Precision-Isotonic”, “Uniasa”, “Guarantee Plus”, and “liquid Pepti-2000 LF”.

In addition, a total of six types of objectives were found: 10 studies sought to compare two different types of EN, among which five of the works compared an elemental formula with a polymeric one, two compared an elemental formula with another elemental one that contained a greater concentration of fats, one work compared two types of polymeric formulas, one work compared two types of elemental formulas, and one work compared an elemental formula with a semi-elemental one.

Moreover, seven studies compared a type of EN with an oral diet, five studies sought to experiment with a type of EN, three studies compared a type of EN with a type of PN, three studies sought to compare a type of EN with another type of medication plus an oral diet, and finally, two studies sought to compare a partial EN plus a diet with an oral diet.

As for the manner of administration of the EN, 15 research studies employed a nasogastric tube, three studies utilized a nasoduodenal catheter, two studies used a nasointestinal catheter, seven studies administered the formula orally, and three did not specify the manner of administration.

The total population analyzed in the research studies found included a total of 1070 individuals with IBD, with 1016 diagnosed with CD and 25 with UC.

The main tools utilized by the researchers to obtain results were scores, biomarkers, and tests to measure the activity of the disease: “Harvey–Bradshaw Index” (HBI), the CDAI, the Van Hees activity index (VHAI), the qualification in the classification of the International Organization of Inflammatory Bowel Disease (IOIBD), the Subjective Global Assessment (SGA), the Truelove and Witts index, the simple clinical index, the Crohn’s disease activity score (CDAS); biomarkers such as CRP, ESR, the white blood cell count (WBC), levels of albumin, pre-albumin, transferrin, hemoglobin, platelet count, total bilirubin, alkaline phosphatase, etc.; and medical tests such as ileocolonoscopy. Specific quality of life questionnaires such as the “Inflammatory Bowel Disease Questionnaire” (IBDQ) were also used as were complementary tests such as urine and feces samples and tests to measure the body’s composition such as anthropometries and bioimpedence.

Table 1 shows the main results schematically, as found in the selected articles. Figure 3 and Figure 4 show the scores obtained by the studies for their methodological quality, according to the Cochrane risk of bias tool. 

### 3.2. Meta-Analysis and Meta-Regression

Only 11 clinical trials had common quality and variables needed to be used in the meta-analysis. These 11 trials worked with a total of 15 groups. The final size of the sample was comprised of 272 individuals, all with CD, to which an EN treatment had been given. The common variables were the CDAI, the CRP, and the ESR, and the co-variables type of nutrition, age, and duration of the intervention. Figure 5 shows the effect size of the use of EN. For the three indicators of disease, the effects were positive when comparing the situation at the start and finish of the treatment with EN independently, if the situation with fixed effects (less probable) or random effects (more acceptable) is considered.

The influence of each study on the results of the meta-analysis are shown on Table 2, considering a model of random effects. Figure 6 shows this influence through the Baujat plot. The numbers shown in the figure correspond to the articles shown in the table in the ID column.

The results show that the articles did not influence the results in the case of the CDAI and the ESR, however, study 6 (D. Royall et al. 1994 with Elemental Nutrition) and to a lesser degree, study 12 (Yun Feng et al. 2013 with Polymeric Nutrition) may compromise the results of the meta-analysis for the CRP. However, the heterogeneity, omitting these works, was 95.1% and 96.6% when compared to the overall 97.2%, therefore, a great influence of the CRP on the meta-analysis could not be determined. 

A funnel plot represents the effects observed in the different studies (*x*-axis), and the standard error (*y*-axis). In the absence of heterogeneity and publication bias, the dots shown in the funnel plot should jointly adopt the aspect of a funnel, with the wider part corresponding to the smaller and more precise studies. A lack of symmetry could be due to this publication bias. The funnel plot is shown in Figure 7, where a lack of symmetry can be observed. Therefore, the non-parametric analysis proposed by Duval and Tweedie to analyze this asymmetry should show a lack of articles, and therefore a publication bias. The results of this non-parametric analysis for the fixed-effects model and the random-effects model are shown in Table 3. These results show a possible publication bias in the three variables studied, if a fixed-effects model is assumed; however, the random-effects models did not show this bias.

With respect to the meta-regression, the results are shown in Table 4. There was a dependence of the CDAI score with the period, losing efficacy in prolonged interventions (*p* < 0.05). The CRP showed better results in the EN when using polymeric formulas that were elemental (*p* < 0.001).

## 4. Discussion

Our results included 30 studies (1070 participants). All trials included had a broad scope and had a very varied methodological and clinical heterogeneity. The variables collected were very diverse, with CDAI, CRP, and ESR being the most common. The sample sizes of the studies included were generally small (*n* < 30), thus, a meta-analysis was needed in order to arrive at better conclusions.

A cure for IBD is not known, however, there is evidence of remission and improvement of the symptoms with EEN, which implies the exclusive consumption of an elemental or polymeric substance for many weeks [50], as shown by many of our results. Despite the lack of correlation between IBDQ and the CDAI, correlations were observed between both indexes starting at week 4 of the treatment. A study that focused on the gall bladder was even found, which showed its improvement after day 36 of treatment administration; therefore, aside from reducing the activity or inducing the remission of the disease, this diet could have beneficial effects on organs related with the digestive system [23,25].

The EN formulas tended to contain macronutrients such as amino acids or simple carbohydrates, along with micronutrients such as vitamins. The proteins, carbohydrates, and fats do not reach the ilium or the colon as they are absorbed in the duodenum and jejunum. As for the amino acids they contain, they were named as elemental formulas if they contained free amino acids, semi-elemental if they contained peptides, and polymeric if they contained whole proteins [51]. Different formulations exist, but the ones that do seem to have a positive effect on the maintenance and remission of the disease are elemental and polymeric diets [10,22]. The efficacy of an EN diet does not depend much on it being elemental or polymeric, as shown by some of our results [42,44], since, a priori, both have the same potential for inducing a remission [32,39,46]. However, the meta-regression conducted indicated that a polymeric diet could decrease the CRP better than an elemental one. Additionally, a distinction could be made between them when looking at the economic burden entailed by the use of one or the other and the acceptability by the patients, meaning that, in the adherence to the dietary treatment, the polymeric ones tend to be more accepted by the patients, as they are better tasting [52,53]. The elemental foods are less tolerated with mouth feeding, and generally require a nasogastric tube, which entails complications and patient discomfort. In contrast, the polymeric EN is more tolerable through the mouth by patients, making it the first option for the ill [33,54].

As for the formulation of the EN, studies have also been conducted on the benefits or not of an EN diet rich in fats as opposed to an elemental EN. Just as in other studies, the results of the clinical trials are controversial. Some studies have demonstrated the beneficial effect of the enteral formula rich in fats [55], while others did not show any effect [56] or less beneficial effects [29]. Despite what has been said, some studies have suggested that an EN high in fats could improve gastrointestinal motility and improve the ilium after an operation [57], reducing damage to the intestinal mucosa barrier and the underlying mechanism that could be associated with its antioxidant action after surgical intervention [58].

EEN, combined with some types of medication such as antibiotics [59], seem to improve the disease’s symptoms. Just as shown by our results, EN combined with other types of pharmaceuticals such as prednisone, corticosteroids, and sulfasalazines show a significantly continuous high rate of remission [41,60]. On the other hand, the combination of EN with steroids does not seem to have significant differences in the probability of a relapse [40], perhaps because the steroids do not address the damage produced in the intestinal mucosa, which is the greatest predictor of complications over time [61,62].

Although the mechanism that nurtures the healing of the mucosa by the EN has not been completely determined as of yet, it has been shown that a polymeric formula was as effective as the Infliximab inhibitor of tumor necrosis factor (TNF)-α, and is higher than the hydrocortisone in the maintenance of the function of the intestinal barrier [63]. This is perhaps the reason why significant differences were not found in a study conducted by Gasull et al. in 2001 that utilized two polymeric EN formulas, with the response being similar in both [30]. Additionally, in another study conducted with two polymeric formulas for five weeks, a significant relationship was not found between the treatment with different EN and the changes produced at the level of the disease’s activity [37].

As for the use of elemental and semi-elemental EN, the results were very similar. For example, Mansuf et al. achieved the clinical remission of 16 patients in four weeks with both formulas, and the reduction of the CRP was significant in both groups [36]. The mechanism of action of the semi-elemental diet could be multifunctional, just like as the elemental one, decreasing the intestinal permeability and thus decreasing the loss of fluid. The semi-elemental diet could also reduce the commensal intestinal bacteria that play a role in intestinal inflammation [64,65]. Thus, the use of these types of diets is advisable, either with the use of an elemental or semi-elemental formula for the management of different gastro-intestinal disorders [66].

In 2013, Yun Feng et al. [24] found significant differences between groups subjected to EEN and EN plus an oral diet, although it is interesting to note how the patients refer to a greater subjective well-being when they take EN together with the oral diet when compared to those who are not treated with supplemented EN, despite the biochemical parameters being very similar [21,49]. Some studies suggest that a partial enteral nutrition supplemented with different diets such as elimination, anti-inflammatory, auto-immune diets, or diets low in FODMAP (Fermentable Oligo-, Di-, Mono-saccharides and Polyols) could be beneficial for UC and CD [25,32], although larger controlled assays are needed to back their use [67]. Even patients who were subjected to EN before their operation experienced benefits, not only in their nutritional state, but also with a reduction of inflammation in their disease [68], with patients also experiencing improvements after said intervention [27].

Historically, EN was used and is used as a complementary nutritional treatment for patients with complicated IBD that leads to worrying malnutrition, thus improving their nutritional state [53]. However, the meta-regression from our study showed an inverse relationship between the period of treatment with EN and the improvement shown through the CDAI, that foreseeably, the patients who are subjected to prolonged EN could stop noticing its benefits.

The EN was utilized as an induction therapy for active IBD [53], but it is important to know which EN formulas can be used to boost their anti-inflammatory effects, as there is evidence that supplementary EN is not sufficient for inducing remission, so that it would have to be used exclusively to be able to obtain its anti-inflammatory effect [69,70]. At present, it is well-established that EEN has a strong anti-inflammatory effect with a reduction in the systemic and mucosa inflammatory parameters in a few days, however, the EEN as a long-term therapy is still a challenge, given its lack of palatability and the lack of data to analyze the efficiency of EEN as a maintenance diet [71].

Diverse studies have shown that clinical remission and healing of the mucosa is possible through different nutritional regimes [72]. As for the debate about which is healthier, EN or PN, our results showed that there are studies in which the effectiveness of both seems to be significantly the same for the improvement of the CDAI [45,48]. Bearing in mind that the dietary antigens could be important stimulants for the immune system of the mucosa, intestinal rest with total parenteral nutrition (TPN) is considered as the main option for achieving this rest and for correcting possible nutritional deficits [73]; however when compared to the EN, it does not seem to provide greater benefits. In fact, in one of the studies included in our review [47], EN was the one that seemed to provide the greatest benefits to the patients and to reduce the costs, personal as well as economic, of the different dietary treatments [74]. 

According to European guidelines, the acceptability and the obligatory compliance of the EN are the greatest obstacles found by different researchers when dealing with EN studies. There are clear differences between the studies shown in terms of healing of the mucosa, and therefore the remission of the activity of the disease, which makes them difficult to compare. What is known, however, is that the EEN is a real alternative to immunosuppressive therapy, which exerts its main therapeutic effect on the microbiota, thus reducing intestinal permeability, enhancing barrier defense, and promoting a reduction of pro-inflammatory cytokines [75,76].

This study is not exempt of limitations. With respect to the systematic review, seven articles were not recovered, so our results could be altered. However, the small variability observed through the meta-analysis implies that these articles could substantially vary the results obtained, and in fact, the sensitivity study that analyzed the publication bias showed little alterations on the effect size as well as its confidence interval. Although the quality of these studies has not been introduced in the meta-regression, all the articles that were utilized with this technique were considered to have sufficient quality, so we do not believe that the quality could introduce bias in the findings.

## 5. Conclusions

EN has been shown to have efficacy for the treatment of CD and is compatible with other medicines. As for the CDAI or the rates of remission, there were no differences between EN and PN. Polymeric formulas, when compared to elemental ones, have shown better results with respect to the CRP. The long-term treatment could dilute the good CDAI results that are obtained at the start of EN treatment. 

## Figures and Tables

**Figure 1 nutrients-11-02657-f001:**
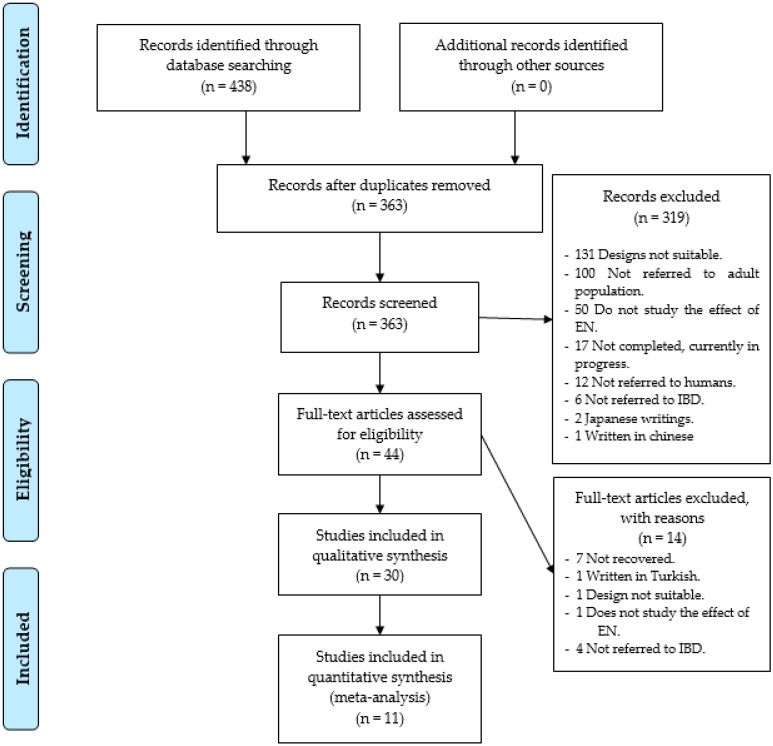
Identification and selection of the studies/records in the databases.

**Figure 2 nutrients-11-02657-f002:**
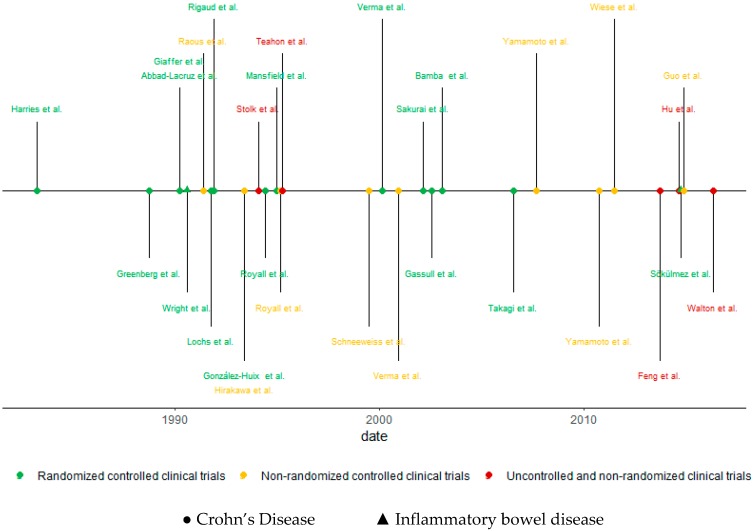
Chronological review according to the type of study and population.

**Figure 3 nutrients-11-02657-f003:**
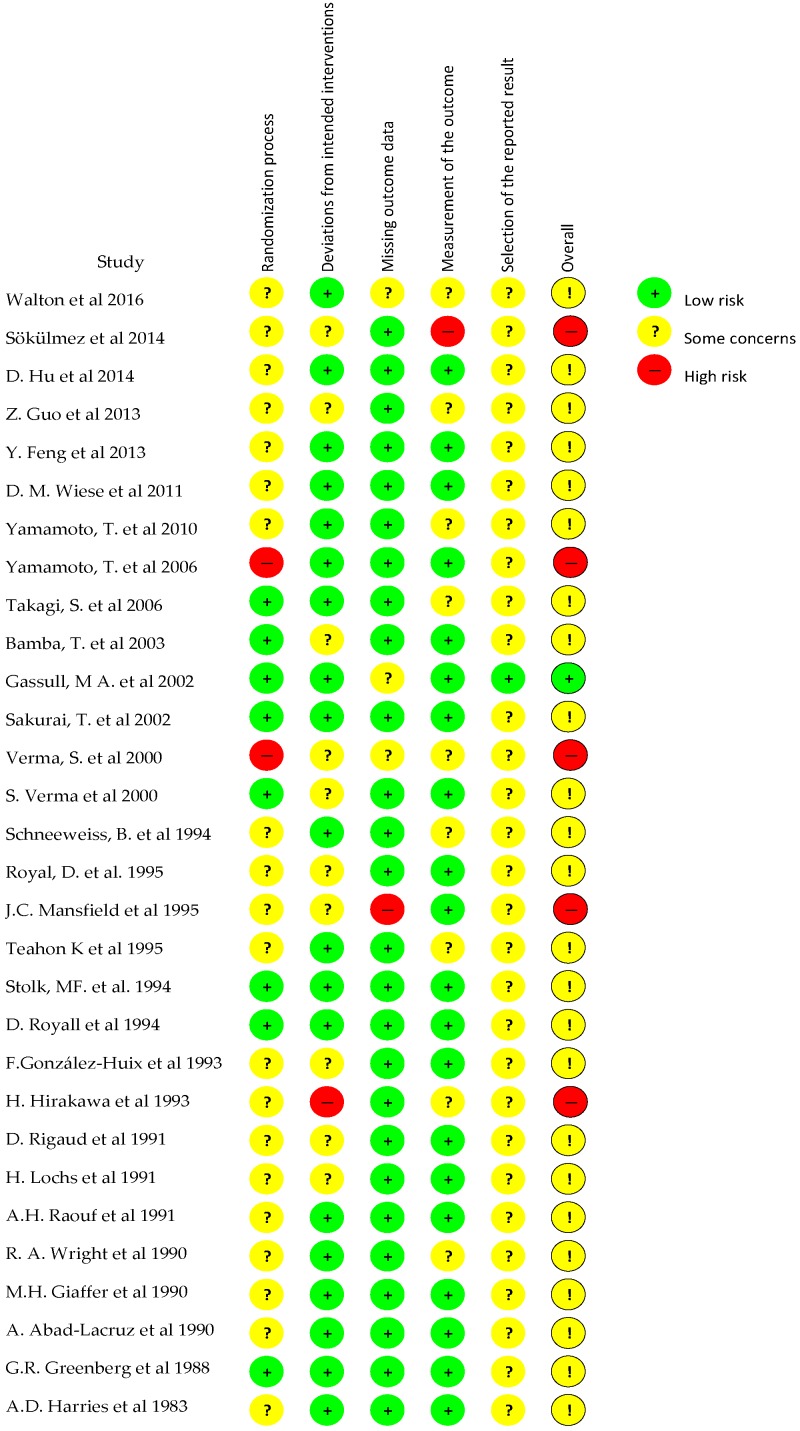
Risk of bias summary across the clinical trials. Low risk of bias: green “+”; Some concerns of bias: yellow “?”, “!”; High risk of bias: red “−”.

**Figure 4 nutrients-11-02657-f004:**
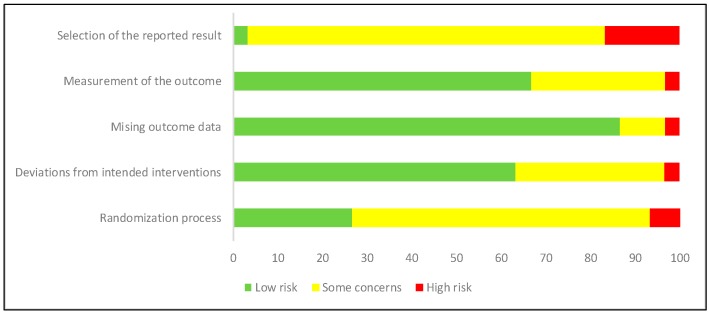
Risk of bias graph across clinical trials. Low risk of bias: green; Some concerns of bias: yellow; High risk of bias: red.

**Figure 5 nutrients-11-02657-f005:**
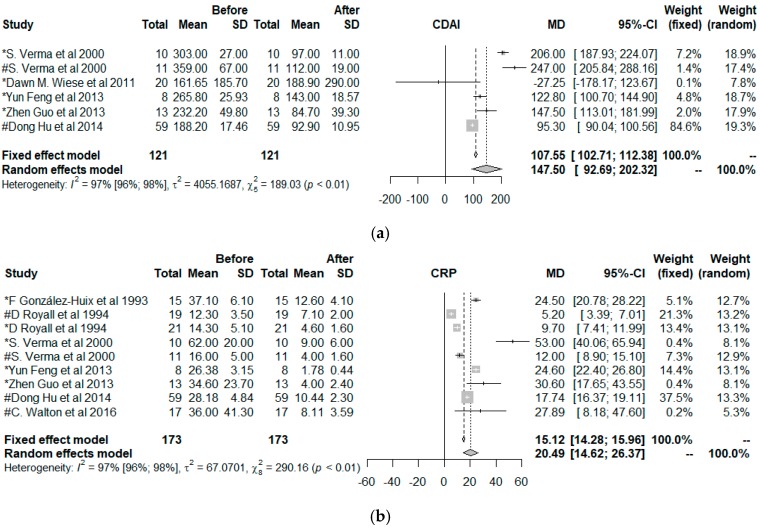
Forest plot for the (**a**) Crohn’s disease activity index (CDAI), (**b**) C-reactive protein (CRP), (**c**) erythrocyte sedimentation rate (ESR). * Polymeric nutrition, # Elemental nutrition.

**Figure 6 nutrients-11-02657-f006:**
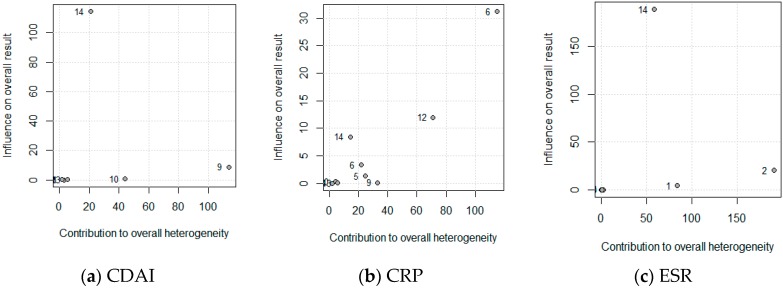
Baujat plot for the (**a**) Crohn’s disease activity index (CDAI), (**b**) C-reactive protein (CRP), and (**c**) erythrocyte sedimentation rate (ESR). The correspondence between the study and the number is shown in Table 2 (ID, Omitting).

**Figure 7 nutrients-11-02657-f007:**
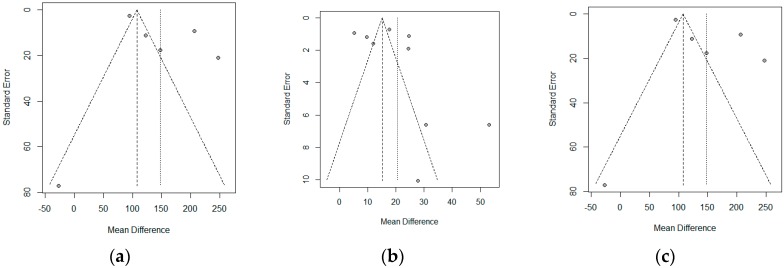
Funnel plot for the (**a**) Crohn’s disease activity index (CDAI), (**b**) C-reactive protein (CRP), and (**c**) erythrocyte sedimentation rate (ESR).

**Table 1 nutrients-11-02657-t001:** Main results of the systematic review.

Author	Study	*n*/age	Disease	P/d	CC	Treatment	Variables	Main Results
Walton et al. 2016 [10]	UNRCT	17/-	ACT CD	14	GB	Enteral feeding E028 extra (Elementary diet)	CRP, HBI and automated spectral identification in feces	The HBI decreased from 6.88 ± 2.93 to 4 ± 5.50, (*p* < 0.05), the CRP from 36.0 ± 41.3 mg/L to 8.11 ± 3.59 (*p* < 0.05), the concentration of 1-propanol and 1-butanol decreased too. No modifications in phenol and indole. The SCFA esters disappeared.
Pinar Sökülmez et al. 2014 [21]	RCCT	38/37 M 28 F 10	ACT IBD EG/CG: 15/23 CD EG/CG: 6/7 UC EG/CG: 9/16	21	TR	EG/CG Diet and EN Novasource^®^/Unrestricted Diet	SGA, BMI, nausea, vomiting, bowel movements, change in malnutrition state, general status, and disease severity.	Although at the beginning of the study the proportion of patients with a severe UC in the EG was higher than in the CG (8/9, and 7/16 respectively), there were no significant differences at the end of the study (*p* > 0.05). In both groups the improvements in disease activity of patients with UC were significant, but non-significant positive changes were observed in the clinical findings during the hospitalization period. Significant improvements of the SGA in both groups.
Dong Hu et al. 2014 [22]	UNRCT	59/32 M 42 F 17	ACT CD	84	CN	Elemental formula Peptide (Nutricia) through nasogastric or nasointestinal tube, plus water and weak tea.	Symptoms, CDAI, peripheral blood samples. Laboratory tests, including nutritional parameters and inflammatory parameters and CT.	50 patients achieved a partial remission, 30 a complete remission. 48 symptomatic remission, 35 radiological remission and 42 clinical remission. The CDAI decreased from 188.2 to 132.4 in 21 days (*p* < 0.05), and to 92.9 after 81 days (*p* < 0.05). Significant decrease in the thickness of the intestinal wall and an increase in the area of the luminal cross section. CRP and ESR decreased significantly (*p* < 0.05), the BMI, albumin, prealbumin and transferrin, HB, platelets, red blood cells, globulin and total protein increased significantly (*p* < 0.05).
Zhen Guo et al. 2013 [23]	UNRCT	13/26 M = 9 F = 4	ACT CD	28	CN	Exclusive EN through polymer formula Administration: Nasogastric tube at night and orally by day. They allowed water consumption.	IBDQ, CDAI, BMI, CRP, ESR, WBC count, HB and serum albumin level in peripheral venous blood.	11 patients achieved clinical remission and 2 did not. CDAI and CRP decreased from 232.2 and 34.6 to 84.7 and 4.0 (*p* ≤ 0.001). Significantly decreased the number of liquid or soft stools, abdominal pain, general well-being and percentage deviation of the standard weight (*p* < 0.05), no differences were found in the presence of complications, taking atropine/diphenoxylate or opiates, presence of a mass abdominal and hematocrit. There were significant improvements in the IBDQ, from 128.3 to 182.9 (*p* < 0.001). Significant improvement in all categories: intestinal symptoms (from 41.5 to 62.0, *p* < 0.001), systemic symptoms (16.5 to 27.5, *p* < 0.001), social function (20.5 to 26.5, *p* = 0.03) and emotional state (49.8 to 66.9, *p* < 0.001). Correlation between IBDQ and CDAI after treatment (r = −0.57; *p* = 0.042).
Feng Y et al. 2013 [24]	NRCCT	24/33 M 17 F 7	ACT CD ENG/NoEN/CG 8/8/8	28	CN	ENG: Enteral formula “Peptisorb” by nasogastric tube, plus water. NoENG: EC patients without EN. CG: Patients with colon carcinoma.	Adipocyte size, adipokine production and level of CRP were evaluated. Leptin, resistin, TNF, and IL-6 and IL-10 levels were determined. BMI, CDAI, etc. were calculated.	ENG patients had a higher BMI level and lower levels of CRP and CDAI (*p* < 0.001) and achieved clinical remission (CDAI < 150). In addition, protein levels of proinflammatory adipokines (TNF-alpha and leptin) were lower, leptin was negatively regulated, and adipokine expression (mRNA level) was positively regulated. In the NoEN group the level of adiponectin protein was higher
Dawn M. Wiese et al. 2011 [25]	NRCCT	20/46 M 4 F 16	ACT CD EPA>2%/EPA<2% 10/10	120	US	Two 8-oz each day of NE EPA>2% or EPA<2% respectively.	CDAI, IBDQ, nutritional status, micronutrient levels, CRP and body composition among others were measured.	EPA > 2% group increased the BMI, fat mass, fat-free mass, IBDQ (+41.4 [23.1, 47.0]; *p* = 0.002) and the CDAI decreased (−47.8 [−65, −37.8]; *p* = 0.05). There were no differences between groups for the rest of the variables studied.
Takayuki Yamamoto et al. 2010 [26]	NRCCT	56/32 M 36 F 20	REM CD EG/CG 32/24	392	JP	EG. Elemental formula “Elental” by nasogastric tube at night and low-fat foods during the day. CG. Unrestricted Diet	WBC, HB, hematocrit, platelet count, ESR, CRP and albumin. CDAI. Symptoms, adverse effects, stool parameters.	The CDAI did not decrease significantly. No differences were observed between the groups. (*p* = 0.51). The cumulative proportion of patients in clinical remission was not significantly different between the groups.
Takayuki Yamamoto et al. 2006 [27]	NRCCT	40/32 M 26 F 14	ACT CD EG/CG 20/20	+365	JP	EG: Elemental formula “Elental” by nasogastric tube at night and low-fat foods during the day. CG. Unrestricted Diet	WBC, HB, platelet count, ESR, CRP and albumin. CDAI and parameters by ileocolonoscopy.	During the year of follow-up, 1 patient of the EG and 7 in the CG developed clinical recurrence (*p* = 0.048). At 6 months, 5 patients of the EG and 8 of the CG developed endoscopic recurrence (odds ratio, 2.0; *p* = 0.50). At 12 months, 6 patients from the EG and 14 from the CG showed endoscopic recurrence (odds ratio, 5.4; *p* = 0.027)
S. Takagi et al. 2006 [28]	RCCT	51/30 M 37 F 14	REM CD EG/CG 26/25	730	JP	EG: Half of calories, elementary diet through a enteral or oral intake and the remaining half by regular meals. CG: Unrestricted Diet	CDAI. Parameters of: feces, symptoms and laboratory tests.	After an average follow-up of 11.9 months, the relapse rate in the EG was significantly lower than in the CG [34.6% vs. 64.0%; Multivariate risk ratio 0.40 (95% CI: 0.16–0.98)]. No significant changes on the rest of the variables
Tadao Bamba et al. 2003 [29]	RCCT	28/28 M 17 F 11	ACT CD Low/ Medium/ High Fat EN 10/10/8	28	JP	LOWG: 6 packages of elemental diet “Elental” and 6 packages of dextrin MEDG: 6 packages of elemental diet “Elemental”, 3 packages of dextrin and 3 packages of dextrin C-1 (dextrin + soybean oil). HIGHG: 6 packages of elemental diet "Elemental" and 6 packages of dextrin C-1. Administration: Nasogastric tube.	IOIBD, inflammatory markers (CRP, ESR) and body weight were recorded at each follow-up.	No differences in body weight gains. The LOWG’s IOIBD was significantly higher than in the MEDG and HIGHG groups (*p* = 0.048) and the CRP lower after the first week. In the MEDG and HIGHG groups the CRP fluctuated during the study. In the LOWG group the ESR decreased, but for the other groups they remained high or increased during the study. Clinical remission was achieved in 8, 4 and 2 patients in the LOWG, MEDG and HIGHG groups respectively. This remission rate is significant if grouped in LOWG vs. MEDG & HIGHG (*p* = 0.046).
M A Gassull et al. 2002 [30]	RCCT	62/29 M 24 F 29	ACT CD PEN1/PEN2/ESTG 20/23/19	28	ES GB DE	PEN 1: Polymeric EN, rich in n9 monounsaturated fatty acids (MUFA) (oleic acid). PEN 2: Polymeric EN rich in n6 polyunsaturated fatty acids (PUFA) (linoleic acid) ESTG (Steroid group): Prednisone.	ESR, CRP, serum fibrinogen, VHAI, CDAI, NRI, serum albumin and grip strength	The intention-to-treat analysis showed that the remission rates were 20%, 52% and 79% for PEN1, PEN2 and ESTG (*p* = 0.001). Withdrawal from treatment, remission rates were 27%, 63% and 79%, respectively (*p* = 0.008). No differences in remission time and changes in activity rates, inflammatory biological parameters, NRIs and nutritional variables.
Toshihiro Sakurai et al. 2002 [31]	RCCT	36/26 M 30 F 6	ACT CD = 36 EDG/TLG 18/18	42	JP	EDG: “Elental” Formula (Ajinomoto Pharma) low in fat. TLG: Twinline Formula (Otsuka Pharma) large amount of medium chain triglycerides Administration: Tube in the duodenum.	CDAI, VHAI, CRP, ESR, levels of: serum albumin, plasma prealbumin, plasma transferrin and retinol binding protein in plasma and triene/tetraeno ratio.	After 2 weeks, serum levels of linoleic acid, an omega 6 fatty acid, decreased significantly in the EDG group. Without significant differences was observed: a short-term remission in 67% in the EDG and 72% in the TLG, a reduction in the CDAI and the VHAI, a normalization of the CRP and an improvement in the ESR and levels serum; albumin, plasma prealbumin, plasma transferrin and plasma retinol binding protein, the linolenic acid levels decreased in both groups.
S. Verma et al. 2000 [32]	NRCCT	39/40 M 12 F 27	REM CD EG/CG 21/28	365	GB	EG: Oral nutritional supplementation with elemental diet “EO28 Extra”, plus normal diet. CG: Unrestricted Diet.	CDAI, inflammatory markers such as CRP, ESR, albumin, HB and platelet count.	The intention-to-treat analysis showed that the remission rates were 48% and 22% for EG AND CG (*p* = 0.0003). Withdrawal from treatment, remission rates were 60%, and 22%, respectively (*p* < 0.00001). Without showing significant differences were observed: a stability of the levels of CDAI and albumin and an increase in BMI. A significant decrease in ESR was observed
Verma S et al. 2000 [33]	RCCT	21/35 M 8 F 13	ACT CD GA/GP 11/10	28	UK	GA: Free amino acids diet. GP: Polymeric diet. Administration: nasogastric tube. Water was allowed.	CDAI, inflammatory markers (CRP, etc.), BMI and body weight.	Clinical remission was achieved in 8 (80%) and 6 (55%) patients in the GA and GP groups, respectively (without significant differences, *p* = 0.1). In both groups CDAI (GA, 359 ± 67 to 112 ± 19, *p* ≤ 0.0002; GP, 303 ± 27 to 97 ± 11, *p* ≤ 0.0005) and CRP (GA, 16 ± 5 to 4 ± 1.6, *p* < 0.1; GP, 62 ± 20 to 9 ± 6, *p* < 0.04) decreased. Remission was achieved earlier in GA (7 ± 2 days) than in GP (14 ± 2 days) (without significant differences). Overall, enteral feeding was successful in 14 patients (63%).
Bruno Schneeweiss et al. 1999 [34]	NRCCT	26/28 M 9 F 17	ACT CD EG/CG 7/19	15	AT	EG: 7 patients received enteral nutrition by nasogastric tube	Energy expenditure, UNP, changes in the body’s urea nitrogen set and body composition.	The REE did not change. From day 7 the UNP, RQ and RQ without proteins increased significantly. These changes (except carbohydrate oxidation rates) were reversed when the EN was interrupted.
Dawna Royall et al. 1995 [35]	NRCCT	60/30 M 32 F 28	ACT CD EG/CG 30/30	21	CA	EG: one of two elementary diets, Peptamen or Vivonex-TEN, administered by nasoduodenal tube.	Total body protein, fat, water and body potassium.	Compared to the CG, the EG lost 11.3 kg (*p* < 0.0005), (5.1 kg fat (*p* < 0.0005), 2.2 kg protein (*p* < 0.025), 3.7 kg water, 24.9 g body potassium (*p* < 0.01)). After EN, body weight (1.9 ± 0.3 kg; *p* < 0.0005), body protein (0.3 ± 0.1 kg; *p* < 0.025), fat (0.3 ± 0.1 kg; *p* < 0.025) and water (1.1 ± more; 0.4 kg; *p* < 0.025) was significantly increased. Body potassium increased but not significantly.
Mansfield JC et al. 1995 [36]	RCCT	44/- M 16 F 28	ACT CD GA/GP 22/22	28	GB	GA: Enteral formula based on amino acids “Elemental 028”. GP: Enteral formula based on oligopeptide-based diet “Pepti-2000 LF liquid”. Water was allowed.	CDAI, laboratory activity measures (HB, platelet count, ESR, serum albumin concentration, AAGP and CRP) and body weight.	16 patients (36.4%) achieved clinical remission and decreased CRP (*p* = 0.05). Both groups had identical rates of remission, failure, early withdrawal and nasogastric feeding intolerance. There was an increase in serum albumin in patients who started the study at a low level.
Teahon K et al. 1995 [37]	UNRCT	19/37 M 10 F 9	ACT CD	35	GB	Elemental diet “Vivonex” was using in one group (*n* = 8) and “Elemental 028” in the other (*n* = 11), by oral route.	CDAS, biochemical parameters (HB, platelet count, leukocytes, ESR, iron, magnesium, copper, zinc…), fecal parameters, BMI and body composition.	Changes were similar in both groups. Clinical disease activity and fecal excretion of leukocytes were significantly reduced after 2 weeks of treatment. Transferrin, prealbumin, albumin and serum iron were significantly increased at 4 weeks. Serum copper decreased during the study period. Changes in nutrition measures did not correlate significantly with changes in disease activity.
M.F.J. Stolk et al. 1994 [38]	UNRCT	6/27 M 3 F 3	CD	42	NL	By using a pump, the formula “Peptison” (Nutricia) was supplied.	Volume, motility, emptying and filling variables of the gallbladder were calculated, and concentration of CCK in the plasma	At the start of treatment, the fasting gallbladder volume decreased from 19.3 +/− 4.5 to 4.9 +/− 3.6 mL. The CCK increased from 1.5 +/− 0.3 to 3.9 +/− 1.1 pmol/L. After 8 days, the gallbladder contracted almost completely, the CCK increased to 7.5 +/− 2.7, and at 36 days, CCK increased to 8.3 +/− 2.6 pmol/L. After 22 days 22 the volume of the gallbladder increased, and after 46 the CCK decreased. This change was significantly greater than the CCK change on day 1 (*p* < 0.05)
D Royall et al. 1994 [39]	RCCT	40/31 M 23 F 17	ACT CD AG/PG 19/21	21	CA	AG: Enteral formula based on amino acids “Vivonex-TEN”. PG: Enteral formula based on peptides “Peptamen”. Administered by nasogastric tube. Water was allowed.	CDAI, CRP, AAGP, phospholipids, albumin and transferrin. Body weight and total body nitrogen was evaluated.	After 21 days, remission rates were equivalent between the two groups: 84% for the AG and 75% for the PG (*p* = 0.38). At 12 months, it remained at 31% and 40% respectively (*p* = 0.39). Also, the reductions of CDAI, AAGP and CRP were significant. Linoleic acid decreased and total body nitrogen increased significantly in AG but not in PG (*p* < 0.025). The concentration of phospholipids in plasma increased significantly in the PG
F González-Huix et al. 1993 [40]	RCCT	32/31 M 17 F15	ACT CD PENG/ESTG 15/17	28	ES	PENG: The polymeric EN administered by nasogastric tube. ESTG: Prednisone administration. And diet lactose-free while they were in the hospital.	VHAI, CRP. Evaluation of body weight, % IBW, MAMC, TSF, serum albumin concentration. Complete hematological and biochemical analysis.	There were no significant differences in the mean time (*p* = 0.47) and the number of patients who obtained clinical remission (*p* = 0.43). The VHAI decreased in both groups; PENG from 172.5 to 113.8, (*p* = 0.0001), ESTG from 184.3 to 118.1, (*p* = 0.0003). In both groups the CRP decreased and the serum albumin concentration increased significantly. After one year, 10 patients (66.6%) in the ESTG and 5 (41.6%) in the PENG relapsed. No differences in the cumulative probability of relapse.
Hiroyuki Hirakawa et al. 1993 [41]	NRCCT	61/25 M 39 F 22	REM CD ENG/ENG+D/DG/CG 25/22/8/6	60	JP	ENG: Elemental EN (“Elental”) through nasoenteral tube. ENG+D: ½ ENG + ½ Low-fat diet and prednisolone DG: Low-fat diet and prednisolone CG: Unrestricted Diet	IOIBD, ESR and CRP	The cumulative rates of continuous remission after 1, 2 and 4 years were in the ENG 94%, 63% and 63%; in the ENG + D 75%, 66% and 66% in the DG 63%; 42% and 0%, and in the CG 50%, 33%. and 0%. The ENG had a higher rate than DG (*p* < 0.05) and CG (*p* < 0.01). The ENG + D had a higher rate than the CG (*p* < 0.05). Patients who received more than 30 kcal of EN showed a higher continuous remission rate (*p* < 0.001).
D Rigaud et al. 1991 [42]	RCCT	30/35 M 18 F 12	ACT CD EENG/PENG 15/15	28	FR	EENG: Elementary enteral formula “Vivonex HN” PENG: “Realmentyl” polymeric formula	CDAI, fecal production, colonoscopies. Body weight; TSF, MAMC, daily urinary, creatinine-height ratio; blood levels of HB, albumin and transferrin. ESR, α2 globulin level and WBC counts.	The clinical remission was in the EENG of 66% and in the PENG of 73%. The CDAI and ESR levels were significantly reduced in both groups. There were no differences between groups for inflammatory markers, colonoscopic lesions, fecal production, body weight and creatinine index.
Herbert Lochs et al. 1991 [43]	RCCT	107/29 M 37 F 70	ACT CD OENG/CSG 55/52	42	DE	OENG: Enteral nutrition by oligopetidic formula “Peptisorb” through nasogastric or nasoduodenal tube. More tea or water. CSG: Combination of corticosteroids and sulfasalazine.	CDAI and laboratory tests.	After 6 weeks, 29 patients achieved remission in the OENG and 41 patients in the CSG (*p* < 0.01). The remission time was significantly different (*p* < 0.01). A CDAI below 150 was achieved in the OENG in 24 patients and in the CSG in 35. The CDAI and severe malnutrition parameters showed no significant differences in patients in remission.
A.H. Raouf et al. 1991 [44]	RCCT	24/-	ACT CD EENG/PENG 13/11	21	GB	EENG: Enteral amino acid-based food “EO28” PENG: Whole protein-based whole food “Triosrbon”. Administration: Oral, flavored with Nesquick.	ESR, erythrocytes, VHAI, Bristol simple activity index and the CRP.	After 3 weeks, they reached remission in the EENG 9 patients and in the PENG 8 patients (*p* < 0.01). The Bristol simple activity index improved in the two groups (EENG; 91.7%, PENG; 86.7% (*p* = 0.35)), Similarly; VHAI (EENG; 18.5%, PENG; 30.0%, (*p* = 0.23)), and CRP (EENG; 58.3%, PENG; 57.1%, (*p* = 0.49)).
Richard A. Wright et al. 1990 [45]	RCCT	11/- M 7 F 4	ACT CD EENG/PNG 6/5	14	US	EENG: Elemental enteral feeding “Vital” PNG: Determined peripheral parenteral nutrition.	CDAI, standard anthropometric parameters, nitrogen balance studies and chemical profiles.	CDAI improved significantly in both groups. Plasma transferrin levels and total lymphocyte count improved in the EENG group (*p* < 0.05). No significant differences in weight gain.
Giaffer MH et al. 1990 [46]	RCCT	30/38 M 8 F 22	ACT CD AG/PG 16/14	28	UK	AG: Amino acid diet “Vivonex”. PG: Polymeric diet “Fortison”. Administration: nasogastric tube. Water was allowed.	CDAI, total body weight, MAMC, TSF and biochemical measurements such as serum albumin.	12 (75%) AG patients achieved remission at 10 days, compared with 5 (35.8%) in the PG group (*p* = 0.03). CDAI decreased significantly in the AG group, not the PG group. The mean weight gain in both groups was similar. Mean serum albumin increased from 26 g/L to 33 g/L (*p* < 0.001). Also, there were significant changes in ESR and AAGP in both groups.
Abad-Lacruz A et al. 1990 [47]	RCCT	22/32 M 15 F 14	ACT IBD PG/TPNG 16/13	NI	ES	PG: Polymeric diet high in nitrogen “UNIASA” by nasogastric tube. TPNG: Specific total parenteral nutrition by a central vein.	Biochemical measurements (total serum bilirubin, alkaline phosphatase, GGT, ALT, and AST) and VHAI and the Truelove and Witts index were measured.	PG had a significant increase in serum albumin concentration (32 ± 1 to 38.2 ± 1.6 g/L; *p* < 0.01). There was lower disease activity in both groups (3.31 ± 0.15 to 2.31 ± 0.24, *p* < 0.05 in GP; and 3.38 ± 0.21 to 2.61 ± 0.27, *p* < 0.05 in TPNG). 8 (5 CD and 3 UC) of 13 patients (61.5%) in the TPNG group developed abnormalities in LFT, while in the PG group only occurred in 1 of 16 patients (6.2%) (*p* = 0.002).
Greenberg GR et al. 1988 [48]	RCCT	51/30 M 25 F 26	ACT CD TPNG/ENG/PPNG 17/19/15	21	CA	TPNG: Total parenteral nutrition, more water, plus daily one ampoule of vitamins. ENG: formula diet “Precision-Isotonic”. PPNG: Unrestricted diet and a partial protein/calorie parenteral nutrition.	CDAI, nutritional assessment and biochemical measurements (hematocrit, blood glucose, serum electrolytes, creatinine, magnesium and albumin).	The average CDAI decreased (*p* < 0.01) with no significant differences between groups. Remission rates to discharge were equivalent among the three groups: 12 patients in TPNG, 11 patients in ENG and 9 patients in PPNG and oral diet (X2 1.42 and 1.15; *p* = n/s). Remission rates of 42% in TPNG, 55% in EN and 56% in PPNG at 12 months were equivalent and not influenced by the type of nutritional support initially administered. At 12 months, 18 patients (35%) required surgery, 17 (34%) were medically treated for relapse, and 16 (31%) had sustained remission.
Harries AD et al. 1983 [49]	RCCT	28/37 M 17 F 11	ACT CD G1/G2 14/14	120	GB	G1: 2 months ordinary diet followed by 2 months supplementation with the non-elementary low-waste formula “Guarantee Plus”. G2: same intervention than G1 with invested order.	Nutritional measurements (height, weight, MAMC and thickness of the skin fold), biochemical measurements (serum prealbumin, serum, red cell folate, creatinine height index, platelets, T lymphocytes, etc.) and urine tests parameters.	The general effect of EN during the 2 months was to increase serum albumin, serum protein and prealbumin levels, creatinine height index and T-lymphocyte count. With EN decreased levels of orosomucoids and serum alkaline phosphatase and its activity (*p* < 0.05) Patients felt better when they received EN, although their monthly symptom scores showed no significant benefit.

P/d: Period (days); CC: ISO country codes; UNRCT: Uncontrolled and non-randomized clinical trial; NRCCT: Non-randomized controlled clinical trials; RCCT: Randomized controlled clinical trials; UCT: Uncontrolled clinical trial; IBD: Inflammatory bowel disease; EG/CG: Experimental and Control Group; UC: Ulcerative colitis; EN: Enteral nutrition; CD: Crohn’s disease; ACT: Active disease; REM: Disease in remission; M: Male; F: Female; CDAI: Crohn’s disease activity index; VHAI: Van Hees activity index; CRP: C-reactive protein; ESR: erythrocyte sedimentation rate; BMI: Body mass index; HBI: Harvey-Bradshaw Index, SCFA: Short chain fatty acid, SGA: Subjective global assessment; WBC: White blood cells, CT: computed tomography exam, HB: Hemoglobin; IBDQ: Inflammatory bowel disease questionnaire; IOIBD: International Organization of Inflammatory Bowel Disease rating; NRI: Nutritional risk index; UNP: Urea Nitrogen appearance rate; RQ: Respiratory quotients; REE: resting energy expenditure; CCK: Cholecystokinin; AAGP: Alpha-1 acid glycoprotein; %IBW: Percentage of ideal body weight; MAMC: Mid-arm muscle circumference; TSF: Triceps skinfold thickness; HEEH: Home elemental enteral hyperalimentation; GGT: γ-glutamyltransferase; ALT: Alanine aminotransferase; AST: Aspartate aminotransferase; LFT: Liver function test; TNF: Tumor necrosis factor; IL: Interleukin; htMAT: hypertrophied messenteric adipose tissue; PTH: Parathyroid hormone; IBDNF: Inflammatory bowel disease nutrition formula; EPA: eicosapentaenoic acid; CDAS: Crohn’s disease activity score.

**Table 2 nutrients-11-02657-t002:** Influence analysis in meta-analysis using leave-one-out method (random effect).

				Meta-Analysis for: Effect Size (%Heterogeneity)		
**ID**	**Omitting**	**KN**	**n**	**CDAI**	**CRP**	**ESR**
1	M.H. Giaffer et al. 1990	Pol	14			13.0 (96.8%)
2	M.H. Giaffer et al. 1990	Elm	16			13.1 (93.7%)
3	D. Rigaud et al. 1991	Pol	15			11.2 (97.6%)
4	D. Rigaud et al. 1991	Elm	15			10.4 (97.6%)
5	F. Glez.-Huix et al. 1993	Pol	15		20.0 (97.4%)	11.1 (97.6%)
6	D. Royall et al. 1994	Elm	19		22.2 (95.1%)	
7	D. Royall et al. 1994	Pol	21		22.3 (97.4%)	
8	Teahon K et al. 1995	Elm	19			11.0 (97.6%)
9	S. Verma et al. 2000	Pol	10	136.9 (94.0%)	17.5 (97.3%)	11.5 (97.6%)
10	S. Verma et al. 2000	Elm	11	128.0 (97.2%)	21.9 (97.6%)	10.9 (97.6%)
11	D. M. Wiese et al. 2011	Pol	20	162.3 (97.8%)		
12	Yun Feng et al. 2013	Pol	8	150.3 (97.9%)	19.7 (96.6%)	
13	Zhen Guo et al. 2013	Pol	13	146.7 (97.8%)	19.6 (97.5%)	10.4 (97.6%)
14	Dong Hu et al. 2014	Elm	59	162.7 (92.6%)	21.5 (97.4%)	9.8 (91.2%)
15	C. Walton et al. 2016	Elm	17		20.1 (97.6%)	
	Pooled estimate			145.7 (97.4%)	20.5 (97.2%)	11.3 (97.4%)

KN: Kind of nutrition; Pol: Polymeric nutrition; Elm: Elemental nutrition; CDAI: Crohn’s disease activity index; VHAI: Van Hees activity index; CRP: C-reactive protein; ESR: erythrocyte sedimentation rate.

**Table 3 nutrients-11-02657-t003:** Number of studies that should be added and the estimated effect size.

	Trim-and-Fill Method				Copas Method	
	Fix Model		Random Model		Random Model	
	Nº Studies	Effect Size Estimated 95%CI	Nº Studies	Effect Size Estimated 95%CI	Nº Studies	Effect Size Estimated 95%CI
CDAI	2	98.9 [43.9;153.8]	0	No Changes	0	No Changes
CRP	3	15.3 [9.7;20.9]	0	No Changes	4	18.0 [12.1;23.9]
ESR	5	19.3 [11.2;27.4]	0	No Changes	0	No Changes

Crohn’s disease activity index (CDAI); C-reactive protein (CRP); and erythrocyte sedimentation rate (ESR).

**Table 4 nutrients-11-02657-t004:** Meta-regression.

Result		Co-Variable	Test of Moderators	
	Intercep	KN *	QM	*p*-Value
**CDAI**	167.9	−33.8	0.289	0.591
**CRP**	13.7	12.6	3.977	<0.001
**ESR**	12.9	−3.0	0.106	0.745
	**Intercep**	**Age**	**QM**	***p*-Value**
**CDAI**	225.5	−2.38	0.203	0.652
**CRP**	52.9	−1.0	0.985	0.321
**ESR**	48.3	−1.1	1.555	0.212
	**Intercep**	**Period**	**QM**	***p*-Value**
**CDAI**	235.5	−1.9	5.662	0.017
**CRP**	21.4	−0.0	0.006	0.941
**ESR**	2.5	0.2	1.551	0.213

KN: Kind of nutrition, * Basis elemental enteral nutrition. Crohn’s disease activity index (CDAI), C-reactive protein (CRP), and erythrocyte sedimentation rate (ESR).

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
