# Peer review of "Enteral Nutrition in Patients with Inflammatory Bowel Disease. Systematic Review, Meta-Analysis, and Meta-Regression"

_nutrients, 2019, doi:10.3390/nu11112657_

Round 1

Reviewer 1 Report

Article entilted: Enteral nutrition in patients with inflammatory 2 bowel disease. Systematic review, Meta-Analysis and 3 Meta-regression, presented to peer review pertains to very difficult and interesting problem – influence of different types of nutritional support and treatment on intestinal condition in IBD. It was a great workload for authors, and all statistical calculations are made properly. However I have two really big remarks concerning assumptions and composition of the trial, that in my opinion disqualify this metaanalysis form publcation. Studies used in metaanalysis (dating from 1988) are really old ones. Another issue is approach to Jadad cassification, in my opinion paper / study reaching 1 (one) point in Jadad scale cannot be assumed as valuable paper. It is widely known, that we are missing studies in this field, we have to wait for new ones, and trying to create new reality from less valuable trials is not a good idea.

Author Response

Comments and Suggestions for Authors

Article entilted: Enteral nutrition in patients with inflammatory 2 bowel disease. Systematic review, Meta-Analysis and 3 Meta-regression, presented to peer review pertains to very difficult and interesting problem – influence of different types of nutritional support and treatment on intestinal condition in IBD.

It was a great workload for authors, and all statistical calculations are made properly.

However I have two really big remarks concerning assumptions and composition of the trial, that in my opinion disqualify this metaanalysis form publication.

Point 1: Studies used in metaanalysis (dating from 1988) are really old ones.

Response: We appreciate your suggestions and really agree with you that some of our results are old. When we began our review of the scientific literature, we tried to focus on the most recent studies, specifically the last 10 years, but the results were few documents. Once we reviewed them, we saw that these referred to previous studies that met our inclusion criteria. Therefore, after a consensus, we agreed to broaden the search date in order to recover the largest number of studies that met our objective. All of them provide knowledge of great value.

Despite of the fact that some studies are old, this is the first Systematic Review and first meta-regression about the issue, and the results help to achieve new knowledge currently unknown.

Point 2  Another issue is approach to Jadad cassification, in my opinion paper / study reaching 1 (one) point in Jadad scale cannot be assumed as valuable paper.

Response: Thank you for this important observation. This point is a weakness of our study, therefore, and following the suggestion of reviewer 2, we have used 'Cochrane risk of bias' tool instead of Jadad classification.

We have edited the manuscript according to the results of the Cochrane risk of bias tool and there are some concerns about the quality of some studies (see the limitations paragraph). However, those included in the meta-analysis did not have a high risk of bias.

Point 3: It is widely known, that we are missing studies in this field, we have to wait for new ones, and trying to create new reality from less valuable trials is not a good idea.

Response: We agree there are few studies in this field and the methodologies of clinical trials have improved over time and now they are more rigorous. However, the Cochrane risk of bias tool does not consider the time as parameter of bias.

Reviewer 2 Report

Comprehensive systematic review with appropriate use of statistics and meta-regression.

Would a subgroup analysis with type of EN diet be useful?

You have acknowledged the limitations of the review w.r.t heterogeneity and non access to 7 articles, were any attempts made to access these articles and also to contact authors for data? Please clarify.

I would recommend using the 'Cochrane risk of bias' tool rather than the Jadad scale for clinical trials, as Jadad doesnt consider allocation concealment and could be over-simplistic.

In the search strategy, please replace 'cinhal' with 'CINAHL"which is the correct database.

Author Response

Comments and Suggestions for Authors

Comprehensive systematic review with appropriate use of statistics and meta-regression.

Point 1 Would a subgroup analysis with type of EN diet be useful?

Response:  We appreciate the reviewer’s suggestion. We preferred to include the type of diet (Elemental or Polymeric) in the meta-regression. In this way, we have a statistical test to check the significant differences between the two types of diets. Doing a meta-analysis in two subgroups does not allow us to test the difference.

Ponit 2  You have acknowledged the limitations of the review w.r.t heterogeneity and non access to 7 articles, were any attempts made to access these articles and also to contact authors for data? Please clarify.

Response: In order to obtain studies that are not accessible via the internet, we have used 3 methods: Researchgate, the correspondence author and the interlibrary loan. Only 3 were recovered through the interlibrary loan.

We have included these facts in the methodology section of the manuscript.

Point 3. I would recommend using the 'Cochrane risk of bias' tool rather than the Jadad scale for clinical trials, as Jadad doesnt consider allocation concealment and could be over-simplistic.

Response:   Thank you for this important observation. This point is a weakness of our study and, as you suggest, we have now used 'Cochrane risk of bias' tool instead of Jadad classification.

We have edited the manuscript according to the results of the Cochrane risk of bias tool and there are some concerns about the quality of some studies (see the limitations paragraph). However, those included in the meta-analysis did not have a high risk of bias.

Point 4. In the search strategy, please replace 'cinhal' with 'CINAHL"which is the correct database.

Response: We have replaced ‘CINAHL’ by ‘cinhal’.